# Amyloid Precursor Protein and Tau Peptides Linked Together Ameliorate Loss of Cognition in an Alzheimer’s Disease Animal Model

**DOI:** 10.3390/ijms241512527

**Published:** 2023-08-07

**Authors:** Ruth Maron, Yaron Vinik, Michael Tsoory, Meir Wilchek, Ruth Arnon

**Affiliations:** 1Department of Immunology & Regenerative Biology, Weizmann Institute of Science, Rehovot 76100, Israel; ruth.maron@weizmann.ac.il; 2Department of Molecular Cell Biology, Weizmann Institute of Science, Rehovot 761001, Israel; yaron.vinik@weizmann.ac.il; 3Department of Veterinary Resources, Weizmann Institute of Science, Rehovot 761001, Israel; michael.tsoory@weizmann.ac.il; 4Department of Biomolecular Science, Weizmann Institute of Science, Rehovot 761001, Israel; meir.wilchek@weizmann.ac.il

**Keywords:** APP and Tau protein, synthetic linked peptides, Alzheimer’s disease (AD), cognitive function, amyloid plaques

## Abstract

The major proteins involved in Alzheimer’s disease (AD) are amyloid precursor protein (APP) and Tau. We demonstrate that APP1 (390–412) and Tau1 (19–34), linked together with either a flexible or a rigid peptide bridge, are able to inhibit, in vitro, the interaction between APP and Tau proteins. Furthermore, nasal administration of biotin-labelled Flex peptide for two weeks indicated the localization of the peptide around and close to plaques in the hippocampus area. In vivo studies in 5xFAD transgenic (Tg) mice, which exhibit plaque load and mild cognitive decline at four months of age, show that nasal administration of the flexible linked peptide reduced amyloid plaque burden. Additionally, nasal treatment with either flexible or rigid linked peptides prevented cognitive function deterioration. A significant treatment effect was achieved when either treatment was initiated at the age of three months, before severe cognitive deficiency is evident, or at five months, when such deficiency is already observed. The nasally treated mice demonstrated a cognitive ability not significantly different from the non-Tg littermate controls. Testing the effect of the flexible peptide by gavage feeding on the cognitive function of 5xFAD Tg mice demonstrated that feeding as well as nasal treatment significantly improves the cognitive ability of Tg mice compared to control PBS-treated mice.

## 1. Introduction

Alzheimer’s disease (AD) is the most common form of age-associated neurodegenerative disorder, clinically characterized by a decline in cognitive function or dementia [1]. Pathologically, it is defined by the accumulation of two main lesions, amyloid (Aβ) plaques and intracellular neurofibrillary tangles (NFTs) [2]. In 2006, evidence was presented that (Aβ) binds to Tau in solution and that this may be a precursor event to AD [3].

The amyloid hypothesis has dominated the AD field and the development of AD therapeutics has focused mostly on removing Aβ from the brain. However, so far there are no positive results in human clinical trials of drugs that can revert or arrest the progression of AD [4]. The amyloid cascade hypothesis states that first Aβ becomes abnormal and this drives pathology and mediates neurodegeneration [5]. There is now a search for other therapeutic targets including anti amyloid and anti Tau interventions [4]. Evaluations of human neuropathological data show that Tau pathology begins about 10 years before Aβ plaques appear. Therefore, it is unlikely that Aβ initiates the cascade of AD [6].

The hypothesis behind our research is based on the findings that an interaction between APP and Tau plays a role in the induction and/or progression of AD [7,8] and a disruption of this association may therapeutically prevent the cognitive ability deterioration seen in AD.

In our previous publication [9], we demonstrated that the association or binding between the two AD-involved proteins, APP and Tau, can be disrupted by a mixture of two peptides APP1 (aa 390–412 HFQKAKERLEAKHRERMSQVMRE) and Tau1 (aa19–34 GLGDRKDQGGYTMHQD). These peptides were predicted to associate with each other using several algorithm programs [10,11], and were actually demonstrated to crosslink to each other by identifying areas in both APP and Tau proteins as possible areas of interaction between the two proteins. Upon analysis of the crosslinked material which was processed for LC-MS/MS, only one crosslink was identified between APP and Tau, between lysine 370 on APP and lysine 387 on Tau. The crosslinked lysine for APP resides very close to the accordingly predicted APP1 (390–412) peptide, as can be seen by the crystal structure of the region [12]. As for Tau, we selected the peptide Tau1 (residues 19–34), which is in the N-terminal end of the Tau protein, since phospho-Tyr-hTau located in the N-terminal was reported to accompany AD progression and Tauopathy [13]. We also selected peptide Tau2 (residues 331–348) from the microtubule area of Tau protein, which is proximal to the crosslinked lysine 387. We tested in addition to APP1 and Tau1 two other peptides as controls for binding to each other both by dot blot and by Eliza testing and found that APP1 and Tau1 Mix were the best candidates for reducing plaques and improving cognitive ability.

A mixture of APP1 and Tau1 peptides administered intranasally to four-month-old transgenic mice (relating to either APP or Tau Tg), exhibiting declined cognitive functions, prevented this decline as well as reduced the formation of amyloid plaques in the brain [9]. The interference with the association of APP and Tau proteins was demonstrated only by a mixture of the two specific peptides and not by either peptide alone. The effect of the peptide mixture on the cognitive functions in the murine AD model 5xFAD mice [14] was attained when treatment was initiated either at an early age (three months), preventing the deterioration of cognition, or at the age of six months, when severe cognitive function loss and plaque load were already evident. These findings, namely improvement in cognitive functions and reduction in the (Aβ) plaque load, indicated the potential of this peptide mixture as therapeutic agent.

The present study sought to evaluate whether a mixture of these two peptides is essential for the reported beneficial effects, or whether the peptides can retain their biological effect when linked to each other via a peptide bridge to yield a single longer peptide. There are two advantages to this bridging approach: firstly, the use of a single defined product is preferable to a mixture of two components. The second advantage is that it can be expressed recombinantly, making it a less expensive procedure than chemical synthesis. In the present study, two versions of such linked peptides were prepared, employing either a “flexible” (3×GGGGS or a “rigid” (3×EAAAK) peptide bridge for the linkage [15,16]. Both versions were consequently evaluated for their beneficial effect in the 5xFAD mouse AD model. To that end, the current study evaluated the treatment effects of the linked peptides on cognitive functions through Y-maze experiments and on plaque load using brain sections labelled with anti Aβ antibodies.

## 2. Results

### 2.1. In Vitro Assessment of Inhibiting APP and Tau Protein Binding by APP1 and Tau1 Peptide Mixture (Mix), or APP1 Linked to Tau1 Either by a Flexible (Flex) or Rigid Linker (Rigid)

Three peptide combinations were used to compare their ability to inhibit the binding of APP and Tau protein, a mixture of APP1 and Tau1 (Mix), and APP1 and Tau1 peptides synthesized with either a flexible linker (Flex) or a rigid linker (Rigid). Their sequences with or without the linkers are described in Figure 1A. To determine if the three combinations have the same or different abilities to inhibit the binding of APP and Tau protein, ELISA assays were performed. Figure 1B shows that APP–Tau binding is inhibited by the mixture of APP1 and Tau1 peptides and that the Flex and Rigid linked peptides had the same or an even more significant inhibitory effect on the binding of the two proteins. This figure represents the results of one experiment, run in triplicate, out of three repeated experiments. We used only one concentration of peptide that was found to be beneficial in all of the other measurements described in this paper.

### 2.2. Capacity of the Flex Peptide to Reach the Hippocampus of 5xFAD Tg Mice

To follow the Flex peptide in vivo, the peptide was synthesized with biotin. 5xFAD Tg mice or non-Tg littermates were nasally treated every other day for a total of seven times, with biotin-Flex (Figure 2A–C). One hour after the last treatment, the mice were sacrificed, their brains excised and verified by staining for plaques with anti Aβ antibodies (6E10) or for biotin with streptavidin followed by anti-mouse antibodies (Section 4). The merge between the plaques and the biotin-Flex with the DAPI staining confirms that the Flex peptide enters the brain and can be identified in the plaque area and around it in the hippocampus (Figure 2A,B). The same treatment in non-Tg littermates, as a control, did not result in any staining of plaques or biotin-Flex (Figure 2C). Some nonspecific staining in blood vessels was seen, but none in the DAPI staining.

### 2.3. Design of In Vivo Experiments Assessing the Effect of Flex and Rigid Peptide in 5xFAD Tg Mice and Non-Tg Littermates

To test the peptide effects in vivo, the cognitive functions of 5xFAD Tg [14] experimental and respective control mice were assessed over several months using the Y-maze test. Mice were tested (Y-maze) before the first treatment (0) and then treated with either PBS (control) or the peptides linked with either a flexible linker (Flex) or a rigid linker (Rigid). Treatment was given three times a week throughout the noted experimental period. Once a month, the mice were tested by Y-maze, assessing spatial recognition memory [17]. An additional cognition test was added, nesting building, to confirm the Y-maze result [18]. The experiment ended by euthanizing the mice and excising their brain for histology.

The first experiment (Figure 3 top) assessed the effects of nasal application of Flex peptide as compared to PBS. The treatment started at the age of three months before symptoms appeared in 5xFAD mice. The mice spatial recognition memory was assessed in the Y-maze after 3, 6 and 8 months of Flex treatment. The second experiment (Figure 3 middle) assessed the effect of nasal application of the Rigid peptide, as compared to PBS, on aged mice. The treatment started at the age of five months, when symptoms typically appear in this line of AD mice. This experiment included an additional control group of non-Tg littermates. The mice spatial recognition memory was assessed in the Y-maze after three and five months of Rigid peptide treatment. The third experiment (Figure 3 bottom) assessed the effect of oral application (by gavage) of the Flex peptide, as compared to PBS. The treatment started at the age of 3–5 months, when symptoms just begin in 5xFAD mice. This experiment also included an additional control of non-Tg littermates treated with PBS. The mice were assessed in the Y-maze after 1, 2, 3 and 5 months of orally applied Flex peptide treatment.

### 2.4. Effects of Nasal Treatment with Flex Peptide on Spatial Recognition Memory and Hippocampal Plaque Load in 5xFAD Tg Mice

5xFAD mice typically begin to demonstrate cognitive impairments at the age of four months [17]. Behavioral assessments were conducted using the Y-maze test assessing spatial recognition memory, as well as a nesting experiment testing cognition functions [18]. Mice were tested before starting the treatment at the age of three months, before cognitive impairment is typically reported in 5xFAD Tg mice. During the treatment period with the Flex peptide, spatial recognition memory was assessed after 3, 6 and 8 months a total of four times (Figure 4A). One-sample *t*-tests indicated that, indeed, at the starting point, both groups displayed intact spatial recognition (PBS; *p* > 0.001; Flex; *p* = 0.039). Two-way repeated measure ANOVA for time, treatment and their interaction demonstrated a significant effect over time (*p* = 0.022). Paired sample *t*-tests indicated that, as compared with pretreatment, a significant decline in Y-maze performance was evident at the 6 and 8 month timepoints among PBS control treatment (6M; *p* = 0.007; 8M; *p* = 0.033), but not among Flex-treated mice (6M; *p* = 0.268; 8M; *p* = 0.758).

Following six months of treatment, an impairment in spatial recognition was evident in both PBS control-treated mice as well as in Flex-treated mice, yet it appeared more pronounced in the PBS-treated mice. However, when tested after two additional months of treatment, the PBS-treated mice continued to exhibit impaired spatial recognition (*p* = 0.733), whereas the Flex-treated mice exhibited intact spatial recognition (*p* = 0.004), and as noted above, did not differ from pretreatment performance at three months of age (Figure 4A).

At the end of the eight-month treatment and behavioral assessment period, their brains were excised and prepared for histology testing (see Section 4). Plaque number and percentage of plaque area, in the hippocampus, were calculated in 5xFAD mice treated with either the mixture of APP1 and Tau1 (Mix) or the Flex peptide which is the two peptides linked together. As Figure 4C depicts, both plaque number and percentage area were significantly and comparably reduced, in the hippocampi of mice treated either with the Mix or the Flex linked peptides as compared to the PBS control-treated group. One-way ANOVA indicated a significant treatment effect for both area percentage (F_(2,12)_ = 33.608; *p* = 0.001) and number (F_(2,12)_ = 9.308; *p* = 0.004). Follow-up planned contrast comparisons (*t*-tests) indicate a significant reduction as compared to PBS-treated mice in area percentage among both Mix (*p* = 0.01) and Flex (*p* = 0.004) -treated mice which did not differ (*p* = 0.458). A similar trend was observed for plaque numbers compared to PBS (Mix: *p* = 0.03; Flex: *p* = 0.034). Mix and Flex-treated mice did not differ (*p* = 0.956).

Mice tested in the nesting experiment were some of the mice tested in the y-maze. The mice were tested at three months of age before starting treatment and tested during treatment after 2, 4 and 6 months (Figure 4D). As compared with pretreatment, a decline in nesting ability was seen in the PBS-treated Tg mice as well as in the Flex-treated mice after four months of treatment. However, when the mice were tested after six months of treatment, the PBS-treated mice continued to decline in their nesting ability, but the Flex-treated mice increased their nesting ability compared to the control (*p* = 0.029) and performed as well as in the pretreatment testing.

### 2.5. Effect of Five Months Nasal Treatment with Rigid Peptide on Spatial Recognition Memory among Aged 5xFAD Tg Mice and Non-Tg Littermates

Behavioral assessments were conducted before starting the treatment at the age of five months when cognitive impairments were already evident. The assessments continued during the treatment period for a total of five months (Figure 5). Y-maze assessments were conducted among 5xFAD Tg mice that were either treated with the Rigid peptide or with PBS control, as well as in a control group of non-Tg littermates. Follow-up one way ANOVA per timepoint indicated no significant main effect for treatment at pretreatment (*p* = 0.816) or three months of treatment (*p* = 0.192). ANOVA yielded a significant main effect for treatment (*p* = 0.034). Planned contrast comparison (*t*-test) indicated that the PBS-treated mice performed worse than both non-Tg (*p* = 0.029) and Rigid-treated mice (*p* = 0.024) that were indistinguishable (*p* = 0.840). However, at five months, a significant treatment main effect was evident (*p* = 0.037). Planned contrast comparison (*t*-test) indicates that the PBS-treated mice performed worse than both non-Tg (*p* = 0.029) and Rigid-treated mice (*p* = 0.024) that were indistinguishable (*p* = 0.840). Moreover, intact spatial retention was evident among aged (10 months old) Rigid-treated mice following five months of treatment (*p* = 0.007) and among non-Tg mice (*p* = 0.010) but not among PBS-treated mice (*p* = 0.877).

### 2.6. The Effect of Oral Administration of Flex Peptide on Spatial Recognition Memory among 5xFAD Tg Mice

In all our previous in vivo experiments, the mice were nasally treated. Considering the future use of our Flex peptide as a possible treatment for humans, we decided to test oral treatment by gavage feeding. 5xFAD Tg mice fed with Flex peptide were assessed for their cognitive functions, as was previously performed with nasally treated mice. The feeding protocol was the same as in the nasal treatment, namely three times a week every other day. However, the dose given by the gavage feeding was 10 times higher based on reports of other peptides or short proteins given orally to mice [19,20]. The monthly follow-up by the Y-maze test, evaluating the memory of the tested mice, was the same as in the nasal treatment. Mice were 3–5 months old at the beginning of the treatment. As shown in Figure 6, already after 60 days of treatment, there was a significant difference between the PBS-fed 5xFAD Tg mice and the mice fed with the Flex peptide. Moreover, performance in the Y-maze of the Flex-treated mice throughout the experiment was indistinguishable from that of the non-Tg littermates and from their assessment before treatment.

As is clearly evident in Figure 6, at the pretreatment timepoint, mice from all three groups exhibited intact spatial retention (non-Tg *p* < 0.001; PBS *p* < 0.001; Flex *p* < 0.002). Repeated measure ANOVA indicated a significant effect for time (F = 3.875, *p* = 0.008). Follow-up *t*-tests indicate a significant decline in spatial retention only in 5xFAD Tg PBS treatment. After two months of treatment, the PBS-treated mice declined significantly compared to pretreated mice (2M *p* = 0.003; 3M *p* = 0.020; 5M *p* < 0.001). This decline led the mice to exhibit impaired spatial recognition that persisted until the fifth month (2M *p* = 0.450; 3M *p* = 0.745; 5M *p* = 0.380). In contrast, orally administered Flex mice exhibited intact spatial recognition (2M *p* > 0.001; 3M *p* = 0.031, 5M *p* = 0.005) that appeared similar to the non-Tg littermates and did not differ from their own before the treatment started (pre-treatment vs. 1M *p* = 0.744; 2M *p* = 0.627; 3M *p* = 0.460; 5M *p* = 0.625).

## 3. Discussion

The current study demonstrates that APP1 (390–412) and Tau1 (19–34) peptides, linked together with either a flexible or a rigid peptide bridge, are able to inhibit, in vitro, the interaction between APP and Tau proteins. Furthermore, these linked peptides concomitantly reduce the Aβ plaque burden in treated 5xFAD Tg mice, a model for AD, and improve their cognitive functions. APP and Tau are involved in AD development, likely by binding to each other. The linked peptides are able to inhibit, in vitro, the interaction between APP and Tau, indicating the therapeutic potential of the linked peptides toward drug development. Additionally, our ~six-month oral treatment with the Flex linked peptide showed a full recovery of cognitive functions, reaching identical levels to those of the non-Tg mice. Taken together, the nasal and oral results strongly indicate the peptides’ potential toward drug development.

In our previous publication [9], we described that a mixture of two peptides (APP1 390–412 and Tau1 19–34) prevents the association of the two brain proteins APP and Tau and improves the cognitive ability in a transgenic mouse model of AD.

In the context of drug development, using a mixture rather than a single component is a complication due to the varying pharmaco-kinetics of the different agents [21]. To avoid this complication, several approaches have been suggested, including the use of nano particles [22]. In the present study, since the two components are peptides, a simpler solution is linking the two peptides into a single longer peptide. There are two advantages to this approach: firstly, the use of a single defined component is preferable to the use of the two-peptide mixture. The second advantage is that it can be expressed recombinantly, a less expensive procedure than chemical synthesis. The linkage could be provided by either a flexible (3×GGGGS) or a rigid (3×EAAAK) peptide bridge [15,16]. It should be noted that, in both versions of the linked peptide, each of the two individual peptides is exposed and available to interact with the other, thus simulating the effect of the mixture of the peptides. Since it was not possible to predict in advance which combined peptide is expected to be more effective, both versions were prepared and evaluated for their biological effect, namely for their effects on plaque burden and cognitive functions in a mouse model of AD.

The delivery procedure is a significant consideration in drug administration. In certain cases, the delivery system is mainly intravenous transfusion (e.g., monoclonal antibodies). However, smaller molecules of various compositions and sizes may be delivered non-invasively (e.g., orally or nasally) [23]. In general, non-invasive drug delivery is the preferred method of delivery and, in AD patients specifically, it may be easier for them to take a pill. In the treatment of multiple sclerosis, there are several drugs in use, some of them administered orally [24] and some by injection [25]. The latter include beta interferon, as well as Copaxone^®^, which is widely used due to its high efficacy and safety profile [26]. Nevertheless, many patients prefer taking orally delivered drugs [27].

The results of the biotin-labelled conjugated peptide delivered nasally show that it indeed reaches the brain of transgenic mice, leading to a substantial reduction in the plaque load. These results are indeed promising for a peptide that is ~50 amino acids long. Furthermore, nasal administration leads not only to delivery of the peptide to its target organ, but also and mainly, to its efficacy in preventing the loss of cognitive functions in the treated mice. It is important to emphasize that no accumulation of this product was observed in similarly treated non-Tg mice, thus providing reassuring safety considerations.

Similarly, and of even higher practical consideration for drug development, is the effectiveness of the linked Flex peptide when delivered orally by gavage treatment. This effectiveness, protecting cognitive functions, was evident following 5–6 months of treatment (every other day, three times per week) that started at the age of three months, before cognition deficiency occurred. Such treatment may be considered as a preventive treatment. It should be noted that, in our previous study, using a mixture of the two individual APP and Tau peptides, we demonstrated a similar protective effect on cognition not only when administrating the Mix at three months of age, before cognitive deficiency is observed, but also at the age of six months, when cognitive impairments were already evident. In both cases, upon completion of the repeated series of treatment, which was continued for 150 days, full cognitive ability was restored [9].

Although the results presented here relate to a transgenic mouse model of AD, they may have implications toward drug development for the treatment of AD in humans, for which no drug is available yet. Hence, the finding that a peptide, which theoretically should not present any safety issue, is efficient in preventing cognitive loss when administered repeatedly, either intranasally or orally, is encouraging. Furthermore, the results of our previous study [9] demonstrate that the effect of the APP1 and Tau1 peptide mixture can be observed not only as a preventive means, but also when initiated at the age of six months, when cognitive impairments are already observed, leading to the recovery of cognitive functions. If pertinent to AD, these combined data may be indicative of possible effectivity, either as preventive, at the very early stages of the disease or at later stages, to alleviate continued cognitive deficiency.

## 4. Materials and Methods

### 4.1. Proteins, Chemicals and Antibodies

Recombinant human APP770, (BLG843201), recombinant human Tau-441 (2N4R) (BLG6842501) and purified anti-amyloid 1–16 6E10 (RRID:RB_2564653) were all purchased from Bio-Legend, San Diego, CA, 92121, USA. Anti-mouse HRP was obtained from Jackson Immunoresearch, Westgrove PA, USA 19390 (Cat#715-035-151). Mouse anti-glyceraldehyde 3-phosphate dehydrogenase antibody (GAPDH) Cat#MAB374 was purchased from Merck-Millipore, MA, USA. Chemicals and antibodies were purchased between 2019–2021.

### 4.2. Mouse Lines

The following mice were used: 5xFAD male and female double transgenic mice (Tg6799 line APP/PS1, JAX: 034848, Jackson, ME, USA), co-expressing the human amyloid precursor protein carrying five familial AD mutations: the Swedish, Florida and London mutations and two mutations of the human presenilin-1. All mouse lines were maintained on a C57Bl/6 background (Jackson, RRID:IMSR JAX:000664). Genotyping was performed by PCR amplification of ear DNA, as previously described [14]. The mice were housed in individually ventilated cages (no more than five mice per cage) in a temperature-controlled facility with a 12-h light/dark cycle. All animal care and experimental use were in accordance with the Weizmann Institute of Science guidelines and were approved by the Weizmann Animal Care Committee IACUC #18691119-2 (31 December 2019–2 January 2023). Animal weight was 20–25 g, and mice were given food and water ad libitum.

### 4.3. Synthetic Peptides

Two synthetically linked peptides were prepared by linking APP1 and Tau1 peptides with a flexible linker or with a rigid linker. Another synthetic peptide of Biotin-flexible peptide was prepared as well, all by GL Biochem Ltd. (Shanghai, China). Custom-made materials can be shared upon reasonable request.

APP1: HFQKAKERLEAKHRERMSQVMRE;Tau1: GLGDRKDQGGYTMHQDFlex: HFQKAKERLEAKHRERMSQVMREGGGGSGGGGSGGGGSGLGDRKDQGGYTMHQDRigid: HFQKAKERLEAKHRERMSQVMREEEAAAKEAAAKEAAAKGLGDRKDQGGYTMHQD

### 4.4. In Vitro Tests

An enzyme-linked immunosorbent assay (ELISA) plate (96 wells, flat bottom) was coated with Tau protein 1 µg/mL bicarbonate PH = 8.2 (50 µL/well), then incubated overnight at 4 °C. At the same time, Flex and Rigid peptides (1 µg/mL PBS) and Mix peptide which is a combination of APP1 and Tau1 (0.5 + 0.5 µg/mL PBS), were also incubated overnight at 4 °C in PBS. The next day, the plate was washed 3× with PBS and blocked with 3% BSA/PBS (10 µL) for two hours at RT. The plate was washed again 3× with PBS, and peptides or their combinations (samples incubated the night before at 4 °C) 50 µL/well were added for four hours at RT. The plate was washed again 3× with PBS, and APP protein, 1 µg/mL per 50 µL/well was added for an ON incubation at 4 °C.

The next day, the plate was washed 3× with PBS, following which anti-Aβ 6E10 (1 mg/mL) was added (1:500 dilution, 50 µL/well) for two hours at RT. The plate was washed again and anti-mouse HRP antibodies (1:10,000 dilution in PBS 1% BSA) were added at 50 µL/well. Color reaction was stopped with 50 µL/well of 1M H2S04 and read at O.D. 450.

### 4.5. Histological Staining and Quantitation of Amyloid

Mice were euthanized with a Pantol injection 20mg/100u, (CTS Israel), and saline-perfused at the end of the experiment as approved by IACUC #18691119-2 (31 December 2019–2 January 2023). The right brain hemisphere of each mouse was stored at −70 °C to quantify Aβ levels. The left hemispheres were fixed (4% paraformaldehyde) for one week, paraffin-embedded and sectioned sagitally (6 microns). Paraffin sections were deparaffinized and rehydrated. Antigen retrieval was performed in 10 mM citric acid pH 6 for 10 min in a microwave to break protein crosslinks and to unmask the antigens. After pre-incubation with 20% normal horse serum and 0.2% Triton X-100 for 90 min, sections were incubated with 6E10 anti Aβ antibody (1:200) at RT for 18hr. The second antibody step was performed by labelling with species highly cross-absorbed cy3 conjugated anti mouse antibody (1:100 Jackson Immune Research, West Grove, PA, USA) for 30–60 min.

For the biotin control animals, the signal was detected by incubation of the sections taken from this group, with cy2 conjugated streptavidin (1:100, Jackson Immuno-Research) for 30 min. All sections were counterstained with Hoechst 33258 (Molecular Probes, Eugene, OR, USA) for nuclear labelling. Stained sections were examined and photographed with a fluorescence microscope (Eclipse Ni-U, Nikon, Tokyo, Japan) equipped with Plan Fluor objectives, connected to camera (DS-Qi1, Nikon).

Quantification analysis of amyloid depositions number, sum of the deposit area and percentage area of deposits (from total measured area), was performed using Image-Pro V-10 software.

The measurements were performed in a fixed rectangle area (3.05 × 1.8) mm, covering all the hippocampus (5.5 mm^2^) in each section, with three sections per animal.

### 4.6. In Vivo Animal Treatment and Behavior Tests

The in vivo research design employed in this study is illustrated in Figure 3. Mice used were heterozygous 5xFAD Tg mice (Tg 6799), and as controls, non-Tg littermates. No predetermined sample calculation was performed and no exclusion criteria were predetermined. A total number of 70 mice were used for the in vivo studies (41 mice in the nasal treatments and 29 mice in the gavage feeding experiment). The mice were kept on a reversed cycle light regime (12–12:lights on at 20:00) to allow for testing during the dark-active diurnal phase. Mice were arbitrarily chosen, but no official randomization method was used. Mice were arbitrarily assigned to experimental treatment groups of peptide treatment or PBS-treated controls (3–9 animals each). No behavioral differences were observed between the sexes when tested in the Y-maze experiment, in terms of response to treatment. Treatment in four different experiments started when the mice were 3 to 5 months old. A mixture of APP1 peptide 5 µg/5 µL and Tau1 peptide 5 µg/5 µL as well as Flex and Rigid peptide (10 µg/10 µL) was administered nasally every other day for 5–7 months. The control 5xFAD-Tg mice, as well as non-Tg littermates, were used as controls, treated with PBS. In the gavage feeding experiment, mice were fed 100ug Flex peptide, diluted in 250 µL PBS per treatment every other day, three times per week. No anesthetics/analgesics were used as neither treatment nor behavioral testing caused discomfort to the mice. The mice were verified three times a week, and no mortality or weight loss occurred. At the end of the experiment, the mice were euthanized by Pantol injection (20 mg/100 µL, CTS Israel) as approved by the ethical protocol, and their brains were excised. One hemisphere was prepared for histology and the other was frozen in −70 °C for processing Aβ 1–42 content.

The Y-maze test [28] consisted of two trials, the first “training” and second “retention”, separated by an inter-trial interval (ITI). Each arm of the Y-maze is equipped with a guillotine door that could be operated manually. The three identical arms were randomly designated as follows: the “start” (steam) arm, in which the mouse began to explore the maze (always open); the “novel” arm, which was closed during the first trial (“training”), but open in the second one (“retention”) and the “other” arm (always open). The first (‘training”) trial lasted five minutes and allowed the mouse to explore only two arms (“start” and “other”) of the maze. Access to the third (“novel”) arm was blocked. The second trial (“retention”) was conducted after a three-minute ITI. During the three-minute trial, all three arms were accessible. The mouse was returned to the same starting arm and was allowed to explore all three arms. Retention was indicated by a preferential index to the “novel” arm, which was calculated as follows: time spent in the “novel” arm minus the time spent in the “other” arm divided by the sum of time spent in both those arms: “0” indicates no preference. Values significantly higher than “0” indicate a preference to the “novel” arm, i.e., adequate retention [28].

We added another behavior test assessing nest building to support the Y-maze results in 5xFAD mice [18]. Mice were placed individually in a cage with two cotton pressed squares. Cages were housed in reversed cycle light regime 12–12: starting from 20 p.m. to 8:00 a.m. We then scored the condition of the nest created using a five-point nest rating scale. All behavioral testing and analysis were conducted in a blinded manner.

### 4.7. Statistical Analysis

Statistical analysis was performed in R, using the rstatix package. For the in vivo studies, a repeated measures ANOVA was performed to assess the significance of the time factor (pre-treatments vs. follow-up measurements), treatment factor (Rigid/Flex/Mix/PBS/non-transgenic) and their interaction. Follow-up post hoc tests were performed by pairwise *t*-tests, where relevant. One-sample *t*-test was used to compare Y-maze results to chance levels (μ=0). A probability value (*p*) of less than 0.05 was considered significant.

## 5. Conclusions

Peptides represent a unique class of pharmaceutical compounds and can be developed to serve as drug candidates to interrupt protein–protein interaction [29]. Some peptides have an advantage over antibodies in cell and BBB permeability. In a previous publication, we demonstrated that the association between APP and Tau can be disrupted by a mixture of two peptides, namely APP1+Tau1 [9]. The purpose of the present study was to evaluate whether a mixture of APP1 and Tau1 peptides is essential for disrupting the association between APP and Tau proteins, or whether they can retain their biological effect when linked to each other to yield a single longer peptide. Two versions of the linked peptides were synthetized either with a rigid bridge or a flexible one. Both versions were evaluated for their beneficial effect in the 5xFAD Tg AD model mice. Both linked peptides showed efficacy in improving the cognitive functions, as well as reducing plaque load, not significantly different than the mixture of the two peptides. 5xFAD Tg mice as well as non-Tg littermates were nasally treated with biotin labelled Flex and verified for the presence of the peptide in the hippocampus. We identified the biotin-Flex in the hippocampus of the transgenic mice, around and close to plaques, which may explain the reduction in plaque number and improved cognition. Non-Tg littermates did not show any biotin labelling in the brain. We did not see any infiltration or aggregation of cells as a sign of inflammation in DAPI staining nor did we observe any mortality of mice during the treatment. We also demonstrate that gavage feeding of the linked peptides improves the cognitive ability of 5xFAD Tg-treated mice. The experiments described, leading to an improvement in the cognitive ability and reduction in plaque load, have a clinical relevance and potential as future therapeutics for AD.

## Figures and Tables

**Figure 1 ijms-24-12527-f001:**
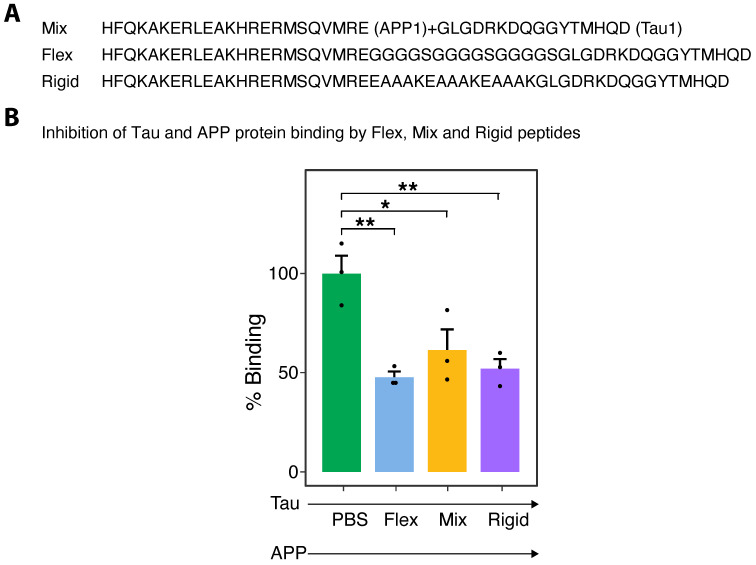
(**A**). Sequence of peptides used for in vitro and in vivo experiments. APP1+Tau1 peptide mixture, the Flexible peptide which is the two peptides connected by a flexible linker (3×GGGGS) or the two peptides connected by a rigid linker (3×EAAAK). (**B**). Binding inhibition of Tau and APP protein by Flex, Mix and Rigid peptides. An ELISA plate was coated with 50 μL/well Tau protein (1 μg/mL) overnight (ON) at 4 °C. PBS, peptide mixture (0.5 + 0.5 μg/mL) and 1 μg/mL (50 μL/well) of the Flex or Rigid peptide were incubated, as well, ON at 4 °C. The plate was washed the next day and blocked with PBS + 3%BSA for two hours at RT. The plate was washed again with PBS. Peptide combinations (Mix, Flex, Rigid) and a control (PBS) were added to the coated plate for four hours at RT. The plate was washed and APP protein was added to all wells for ON incubation at 4 °C. The plate was then washed with PBS and anti Aβ was added to test the ability of the different samples to affect APP protein binding to the Tau-coated plate. The plate was developed with anti-mouse HRP (Section 4). Samples were in triplicates (dots represent the individual repeats). Significance was labelled as * *p*-value < 0.05, or ** *p*-value < 0.01.

**Figure 2 ijms-24-12527-f002:**
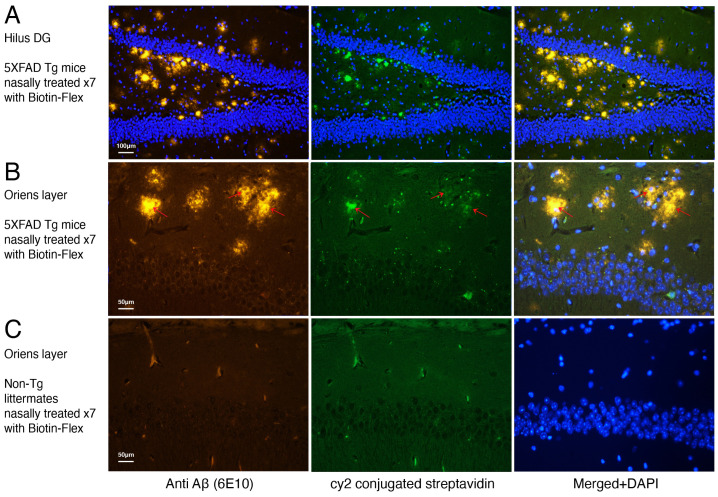
Biotin labeled Flex peptide in the hippocampus of 5xFAD Tg mice. (**A**,**B**). 5xFAD Tg mice were nasally treated with biotin-Flex every other day for a total of seven times. One hour after the last treatment, the mice were sacrificed, their brains excised and stained for plaques (anti Aβ antibody (6E10)) or streptavidin followed by anti-mouse antibodies. The presence of plaques was verified by the 6E10 labelling (in red), and the presence of biotin-Flex was verified by streptavidin labelling (In green). DAPI was used to stain nucleus (in blue). The plaques and the biotin Flex labelling can be seen in the hilus dentate gyrus (DG) area of the hippocampus as well as in the oriens layer of the hippocampus (Merged DAPI panel). Colocaliztion of plaques and FLEX biotin is indicated by red arrows in (**B**). (**C**). To make sure that our results are specific, we treated as controls non-Tg littermates in the same way we treated the Tg mice, as well as the labelling of the brains. As can be seen in (**C**), there was no labelling of plaques or biotin-Flex as expected from non-Tg mice. There is some non-specific labelling in blood vessels which is not seen in the merged DAP1 panel. The merge of plaque and biotin-Flex labelling with DAPI was only identified in (**A**,**B**). However, in the non-Tg mice (**C**), there was only DAPI labelling. Scale bar represents 100 μm in (**A**), and 50 μm in (**B**,**C**).

**Figure 3 ijms-24-12527-f003:**
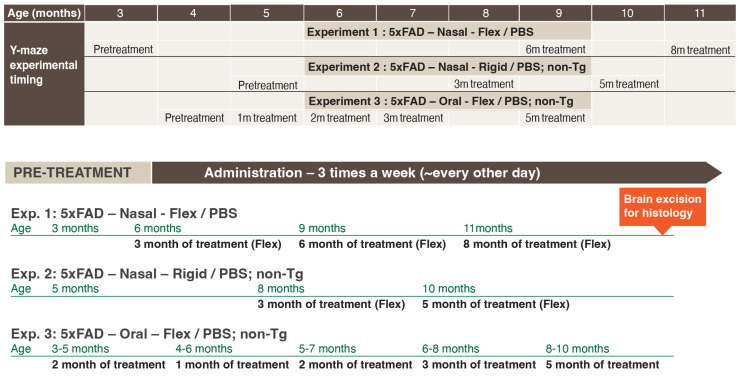
Schematic presentation of in vivo experiments in 5xFAD Tg mice and non-Tg littermates. Three experiments assessed the effects of prolonged chronic administration of the Flex and/or Rigid peptides on spatial recognition in the Y-maze. Exp. 1 (**top panel**) assessed the effects of nasal treatment with Flex peptide over eight months on spatial recognition memory and hippocampal plaque load in 5xFAD Tg mice. Exp. 2 (**middle panel**) assessed the effects of five months of nasal treatment with Rigid peptide on spatial recognition memory among aged 5xFAD Tg mice. Exp. 3 (**bottom panel**) assessed the effects of five months oral administration of Flex peptide on spatial recognition memory among 5xFAD Tg mice.

**Figure 4 ijms-24-12527-f004:**
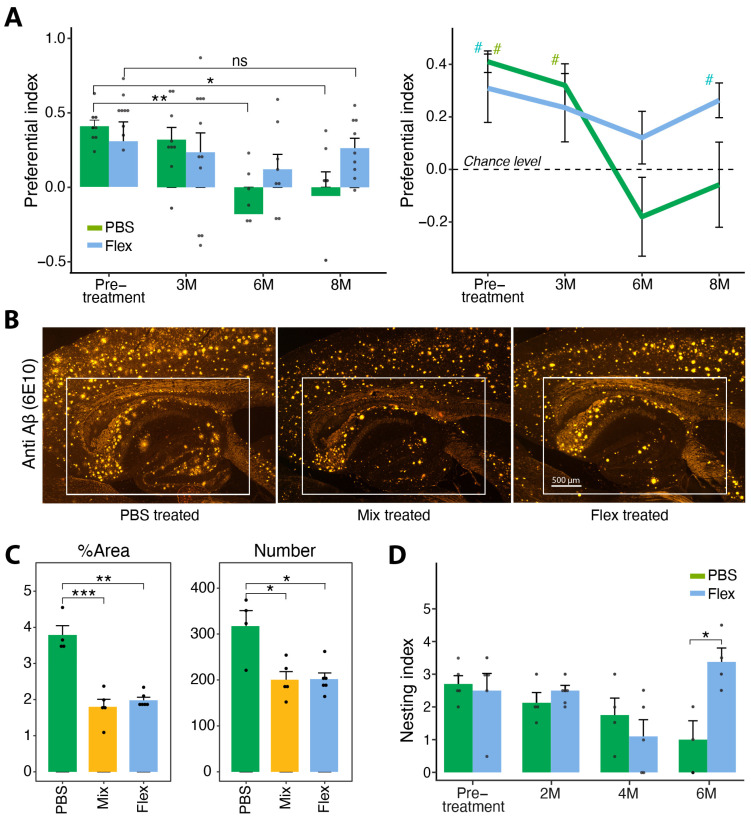
Nasal treatment with Flex peptide conserved spatial recognition memory, nesting ability and reduced hippocampal plaque load in 5xFAD Tg mice. (**A**). Control mice were treated with PBS; drug tested mice were treated with the Mix or Flex peptides. At the end of the experiment, the mice were sacrificed and their brains excised and prepared for histology. */** = *p* < 0.05/0.01] within subjects effects; # = *p* < 0.05 different from chance level (0). (**B**). Staining of plaques with anti Aβ (6E10) in the hippocampus of 5xFAD Tg mice treated with PBS, Mix or Flex for eight months. Control PBS mice have the most plaques in the hippocampus/cortex area, while Mix and Flex-treated mice have a significantly lower plaque number. Scale bar represents 500 μm (**C**). Plaque number (right panel) and % plaque area (left panel) in the hippocampi, collected at the end of the experiment (eight months) from PBS control, Mix and Flex nasally treated mice. Both % area [*p* = 0.001] and plaque number [*p* = 0.004] were significantly and comparably reduced in the hippocampi of mice treated with either Mix or Flex linked peptides as compared to the PBS control-treated group. */**/*** = *p* <0.05/0.01/0.001 within subjects effects. (**D**). As an additional behavior test, we verified 10 male 5xFAD Tg mice for their nesting ability. Five mice were nasally treated with PBS and five mice with Flex. Treatment started at three months and mice were tested at 2, 4 and 6 months of treatment. After 6 months, the Flex-treated mice showed an equal nesting ability to the pretreated three-month-old mice. We analyzed 3–5 males and 4–5 females per group, for a total of 17 mice. Black dots repreprsent individual mice.

**Figure 5 ijms-24-12527-f005:**
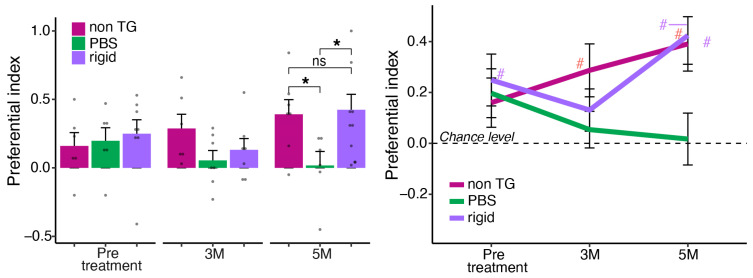
Nasal treatment with Rigid peptide over five months ameliorates spatial recognition memory in 5xFAD Tg mice. Five-month assessment of cognitive function by Y-maze among control PBS nasally treated Tg mice compared to 5xFAD Tg mice treated with the Rigid peptide as well as non-Tg mice treated with PBS. * = *p* < 0.05 within subjects effects; # = *p* < 0.05 different from chance level (0). Twenty-four mice were analyzed in this experiment, 6–9/group male and female. Black dots represent individual mice.

**Figure 6 ijms-24-12527-f006:**
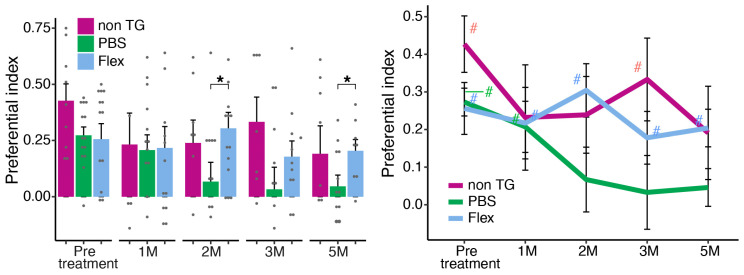
Y-maze performance across treatment groups and throughout the course of treatment. Treatment by gavage feeding of Flex peptide protected 5xFAD Tg mice from age-associated cognitive decline. 5xFAD Tg and non-Tg mice were first tested in the Y-maze before treatment began at 3–5 months old. The feeding protocol (100 μg/250 μL) of the Flex peptide three times a week (every other day) was the same as in the nasal treatment in previous experiments. * = *p* < 0.05 within subjects effects; # = *p* < 0.05 different from chance level (0). We analyzed 3–8 males and 6–12 females per group, for a total of 29 mice. Black dots represent the individual mice.

## Data Availability

The data presented in this study are available on request from the corresponding author.

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
