# Peer review of "Amyloid Precursor Protein and Tau Peptides Linked Together Ameliorate Loss of Cognition in an Alzheimer’s Disease Animal Model"

_ijms, 2023, doi:10.3390/ijms241512527_

Round 1

Reviewer 1 Report

The present research article by Maron et al. entitled “APP and Tau Peptides Linked Together Ameliorate Loss of Cognition in an Alzheimer’s Disease Animal Model” demonstrates the APP1 (390-412) and Tau1 (19-34), linked peptide inhibits in vitro, the interaction between APP and Tau proteins. Two weeks chronic nasal administration of biotin-labelled Flex peptide trigger localization of the peptide around and close to plaques in the hippocampus area. Additionally, nasal administration of the flexible linked peptide reduces amyloid plaque burden and prevents the deterioration of cognitive functions in 5xFAD transgenic (Tg) mice. There are few queries and comments that need to be addressed by authors, as follows:

Query#1; During the prolonged chronic nasal administration of peptides, did authors observe any inflammatory response in hippocampus/frontal cortex? Authors need to show data on microglial/astrocytic activation. Also, authors need to show neurotoxic effect of peptides during chronic nasal administration in 5xFAD Tg mice. This can be achieved by counting hippocampal neurons and normalizing to cortical neurons.  

Query#2; In figure 4, authors need to present high resolution confocal images to show localization of plaques in specific cell populations or brain subregions.  

Query#3; Authors used only one behavioral paradigm i.e. Y-maze to assess spatial memory in 5xFAD Tg mice with/without peptides treatment. However, additional paradigms such as novel object recognition, water maze etc. should be performed for hippocampal spatial memory task.

Query#4; Almost all figures resolution quality is poor. Authors need to revise the figures with better resolution.

English language is fine.

Author Response

       Reviewer 1: Answers to Review report

  • The paper was edited to improve the English

  • The conclusions are supported by the results and a paragraph was added to the conclusions describing the presence of biotin-Flex, in the hippocampus, around and close to plaques which could explain the reduction of plaques and improvement of cognition.

Answers to suggestions

Query #1: Looking at the DAPI staining (Fig. 2) we could not see any infiltration or aggregation of cells which is a sign that there is no evident inflammation. Even after 6-8 months of treatment no mice lost weight or died. Experiments to check if there is a neurotoxic effect as a result of chronic nasal administration and localization of plaques in specific cell populations of brain sub-regions are planned for future studies.

Query#2: We cannot present high resolution confocal images. In Fig.4B, we increased the contrast so the plaques can be seen clearer.

Query #3: While testing the nasal Flex treated mice for behavioral paradigm by Y-maze we also tested a group of ten 5xFAD Tg males treated with Flex or PBS for nest building. The results were added to Fig 4D. We obtained a similar result for nest building as with the Y-maze test.

Query #4: The resolution enclosed in the figures is 1000 pixels and 300 dpi in PDF format. Please see the enclosed PDF file.

Reviewer 2 Report

The Manuscript by Maron et al. titled "APP and Tau Peptides Linked Together Ameliorate Loss of Cognition in an Alzheimer’s Disease Animal Model" aimed to demonstrate the effect of APP1 (390-412) and Tau1 (19-34) linked together with a flexible or a rigid peptide bridge on cognitive function in 5xFAD transgenic (Tg) mice. Authors showed that the feeding as well as nasal treatment with either the flexible or the rigid linked peptides, prevent the deterioration of cognitive functions in 5xFAD transgenic (Tg) mice. The work is interesting. However, need more experiments to support the conclusion of the manuscript. I have some comments as follows:

- In figure 4B, the Authors did not add the scale bar for the images. Moreover, authors should provide high-magnification images.

- The experiments performed are not sufficient to support the conclusion of the manuscript. For example, The authors used Y-maze for the assessment of cognitive function. To support these results, authors should perform more tests including the Morris water maze test for cognitive function assessment.

- The mechanistic understanding of how flexible peptidesameliorate the loss of cognition is not evaluated in this manuscript.

-  Why did the authors select APP1 (390-412) and Tau1 (19-34) regions? Why not others in these proteins? The authors should discuss this more in the manuscript.

Author Response

Reviewer 2: Answers to review report

  1. More background information about the amyloid and Tau hypothesis was added

to  the introduction to support our therapeutic approach for AD as described in the paper.

  1.  Yes, the references cited are relevant to the paper.
  2. We improved our research design (Fig.3) to make it easier to follow.
  3. We went over the methods which are detailed in the Materials and Method section

                 as  well as in the figure legends.

  1. We replaced Table 1 with a figure (Fig. 2) which improve the results presentation.
  2. We added to the conclusion section our biotin-Flex results (Fig.2) showing

                the  entry of the labelled Flex peptide to the brain accumulating around the plaques,

                which supports our conclusions regarding the reduction in plaques and improved

                cognition.

Answers to suggestions

Query #1: Scale bar was added to Fig.4B. We also increased the contrast in Fig.4B so the plaques can be seen clearly.

Query #2: In order to support the results obtained by the Y- maze test used  for behavioral paradigm, in the nasally treated Flex mice, we also tested ten 5xFAD Tg males treated with Flex or PBS for nest building.  Nesting is another assessment test for cognition function and we obtained a similar result in nest building ability as with the Y-maze test. We added the results to Fig 4D.

Query #3: The flexible peptide seems to function similar to the mix treatment described in our previous paper. The Flex peptide is able to bend in any direction and therefore both the APP peptide or the Tau peptide can bind to either of the proteins and block their binding to each other. The hypothesis behind our work is that inhibiting the binding between APP and Tau will block the development of Alzheimer’s. We added our mechanistic understanding to the discussion.

 Query #4: In our previously published paper we described the rational of choosing  APP1 and Tau1 peptides. Using the UNIPROT and BLASTP  programs, we identified areas in both the APP and Tau proteins as possible areas of interaction. Upon analysis of the crosslinked material which was processed for LC-MS/MS, only one crosslink was identified between APP and Tau, between lysine 370 on APP and lysine 387 on Tau. The crosslinked lysine for APP resides very close to the accordingly predicted APP1 (390–412) peptide, as can be seen by a crystal structure of the region. As for Tau, we selected the peptide Tau1 (residues 19–34) which is in the N-terminal end of Tau protein, since phospho-Tyr-hTau located in the N-terminal was reported to accompany AD progression and Tauopathy.

We also selected peptide Tau2 (residues 331–348) from the microtubule area of Tau protein, which is proximal to the crosslinked lysine 387.

We tested in addition to APP1 and Tau1 two other peptides as controls for binding to each other both by dot blot and by Eliza testing and found that APP1 Tau1 Mix were the best candidates for reducing plaques and improving cognitive ability. We added this information to the introduction.

Reviewer 3 Report

In the manuscript submitted by Maron et. al. titled as ‘APP and Tau Peptides Linked Together Ameliorate Loss of Cognition in an Alzheimer’s Disease Animal Model, the authors found the nasal delivery or the oral gavage of the linked APP and Tau peptides before and from the early stage of the amyloid pathology rescued the cognitive deficits in the 5xFAD mice. With the application of Flex-linked peptides, the amyloid burden was also reduced.

Overall, this paper tested the outcomes of ‘linked’ types of APP and Tau peptides and different delivery methods, leading to a story of novel and significance to provide experimental evidence for the potential therapeutic consideration. The paper itself is well written and organized with clear logic. Still, I have some minor comments below.

1. Please double check the reference number for the 5xFAD mouse used since 032284 is not found in Jackson lab website.

2. Line 169-170, please cite an appropriate paper for the statement ‘he Y-maze test assessing spatial recognition memory’.

3. Please clarify the mouse numbers/sex/group used in each experiment in each figure legend. 

Author Response

Reviewer 3: Answers to review report

no further comments necessary

We improved Fig.3 describing the research design as to make it easier to follow. We improved Fig. 4B so the plaques are seen better and we changed Table 1 to Fig.2 so the biotin Flex can be seen entering the hippocampus.

Answers to suggestions.

Query #1: We corrected the 5xFAD Jackson reference number.

Query #2: Reference #28 cites the Y-maze test as appropriate for assessing spatial recognition memory and we also added results obtained in a nesting experiment as an additional test for cognition assessment (Fig 4D).

Reviewer 3: Answers to review report

We improved Fig.3 describing the research design as to make it easier to follow. We improved Fig. 4B so the plaques are seen better and we changed Table 1 to Fig.2 so the biotin Flex can be seen entering the hippocampus.

Answers to suggestions.

Query #1: We corrected the 5xFAD Jackson reference number.

Query #2: Reference #28 cites the Y-maze test as appropriate for assessing spatial recognition memory and we also added results obtained in a nesting experiment as an additional test for cognition assessment (Fig 4D).

Query #3: The number and sex for each experiment was added to each figure legend.

Reviewer 4 Report

Manuscript ID: ijms-2367497
Type of manuscript: Original
Title:
APP and Tau Peptides Linked Together Ameliorate Loss of 2 Cognition in an Alzheimer’s Disease Animal Model

Although the topics and the idea of carrying out the research are interesting and certainly extremely important, I have some very serious reservations. I'm already leaving aside numerous editorial errors such as redundant periods and the equals sign, parentheses once square and once regular, missing verbs in sentences, persistently spelled strep avidin instead of streptavidin, and so on. Although, of course, this also makes the text more difficult to read.

-          I was completely baffled by the absence of Table 2, which the authors describe extensively in the text.

-          Graphically presented evidence that intranasally administered biotin-labeled peptide reaches the brain is lacking.

-          Figure 3 is vague and the description underneath it doesn't make it any easier to understand the experimental scheme either. Authors write about three panels, while I see only 2 panels.

-          I think that with so many experimental groups of animals, there should be information about the size of each group in the figure captions.  

-          Check all the p values in brackets, they seem strange: p=0194 or p=840.

-          Please, someone read this text carefully and correct any inadequacies.

Author Response

Reviewer 4: Answers to review report

  1. We added more information and references regarding the amyloid and Tau hypothesis in the introduction in order to better explain the reason for our research
  2. All cited references are relevant to the research.
  3. Figure 3 describing the research design was improved as to make it easier to follow.
  4. The methods are described in the Material and Method section as well as detailed in the figure legends.
  5. The results are clearly presented and we changed Table 1 to Figure 2 as to

show the results of the biotin peptide entering the hippocampus.

Query #1: The editorial errors were corrected.

Query #2: Table 1 was replaced by Fig.2 which graphically presents evidence that intranasally administered biotin-labeled Flex peptide reaches the brain.

Query #3: We corrected the presentation of Fig.3 so it is easier to follow the experimental plan.

Query 4: The size and sex of each experimental group was added to each figure legend.

Query 5: The p values were corrected to p=0.194 and p=0.840.

Query 6: We read the text as carefully as possible and corrected any inadequacies.

Round 2

Reviewer 1 Report

Authors have replied to the concerned queries, however additional experiments/data are still warranted for drawing a conclusion from this study.

English quality has been improved by authors in this revised manuscript. 

Author Response

we improved the conclusions with more data

 5XFAD Tg mice as well as non-Tg littermates were nasally treated with biotin labelled Flex and checked for the presence of the peptide in the hippocampus. We identified the biotin-Flex in the hippocampus of the transgenic mice, around and close to plaques, which may explain the reduction in plaque number and improved cognition. Non-Tg littermates did not show any biotin labelling in the brain. We did not see any infiltration or aggregation of cells, as a sign of inflammation, in DAPI staining nor did we observe any mortality of mice during the treatment.

Reviewer 2 Report

I think that the authors have addressed all the comments in the revised version of the manuscript. Therefore, I have no further comments.

Author Response

no further comments were needed

Reviewer 4 Report

I have no further comments.

Author Response

no further comments needed